# The pace of shifting seasons in lakes

R. Iestyn Woolway [1] ✉

Lake ecosystems are vulnerable to seasonal thermal cues, with subtle alterations in the timing of seasonal temperatures having a dramatic influence on aquatic species. Here, a measure of seasonal change in temperature is used to describe the pace of shifting seasons in lakes. Since 1980 spring and summer temperatures in Northern Hemisphere lakes have arrived earlier (2.0- and 4.3-days decade$^{-1}$, respectively), whilst the arrival of autumn has been delayed (1.5-days decade$^{-1}$) and the summer season lengthened (5.6-days decade$^{-1}$). This century, under a high-greenhouse-gas-emission scenario, current spring and summer temperatures will arrive even earlier (3.3- and 8.3-days decade$^{-1}$, respectively), autumn temperatures will arrive later (3.1-days decade$^{-1}$), and the summer season will lengthen further (12.1-days decade$^{-1}$). These seasonal alterations will be much slower under a low-greenhouse-gas-emission scenario. Changes in seasonal temperatures will benefit some species, by prolonging the growing season, but negatively impact others, by leading to phenological mismatches in critical activities.

There is compelling evidence that climate change is leading to a major shift in the timing of the seasons over land and at the surface of the ocean[1–7]. Current estimates suggest that spring is occurring earlier, autumn occurring later, and the summer season is lengthening at alarming rates[6–10]. Indeed, shifts in the timing of the seasons are one of the most conspicuous impacts of climate change, which can increase the risk of severe and in some cases irreversible ecological impacts[11]. Whilst much is known about the pace of the shifting seasons in both marine and terrestrial ecosystems, we know very little about these climatic shifts in lakes and how they might respond to future climate change. Indeed, while previous studies have highlighted the long-term response of lakes to a changing climate, particularly relating to their thermal environment, as well as the ecological implications of such changes[12–21], the pace of seasonal shifts has not yet been investigated. This knowledge gap is of considerable concern given the high vulnerability of lakes, and the many ecosystem services that they provide, to climate change. For example, reproductive events of many temperate fish species are influenced by seasonal thermal cues and increasing water temperatures can shift the timing of these events earlier in the spring or later in the autumn[22–25]. These seasonal shifts could also cause a trophic mismatch in aquatic food webs if different trophic levels respond to sub-seasonal temperature cues that change at different rates[26], which is also evident in the oceans[27]. Moreover, the succession of phytoplankton communities responds to seasonal water temperatures[28,29], and these changes may exacerbate water quality

issues, such as the growth of toxic, bloom-forming cyanobacteria[30,31]. Quantifying current changes in the arrival of typical seasonal climates, and how these might change this century, is thus of critical importance.

Here, I quantify past changes and assess future ones in the arrival of typical spring (March–May in Northern Hemisphere and September–November in the Southern Hemisphere), summer (June–August in Northern Hemisphere and December–February in the Southern Hemisphere), and autumn (September–November in the Northern Hemisphere and March–May in Southern Hemisphere) temperatures in lakes worldwide. Following the methods of ref. 5., the pace of shifting seasons in lakes is calculated, for the season of interest, by dividing the long-term trends in monthly surface water temperature (°C year$^{-1}$) by the climatological seasonal rate of change (°C month$^{-1}$)— see "Methods" and Fig. S1. To investigate the pace of the shifting seasons in lakes I focus on three distinct periods: (i) the historic to contemporary period (1980–2021), (ii) the satellite data-taking period (1995–2021), and (iii) the future (2021–2099). For each of these time periods, data from different sources are analysed, as described briefly below (see "Methods" for more information). To quantify seasonal shifts during the historic to contemporary period (1980–2021), simulated surface water temperatures from >150,000 representative lakes worldwide are investigated, with each lake representing an aggregated 'typical lake' for a 0.25° longitude-latitude grid. Specifically, an aggregated lake for each 0.25° grid represents the average lake thermal

[1]School of Ocean Sciences, Bangor University, Menai Bridge, Anglesey, Wales. ✉e-mail: iestyn.woolway@bangor.ac.uk

environment in that location using the grid cell's climate forcing as input to a numerical process-based lake model. These simulations are available from the European Centre for Medium Range Weather Forecasts' ERA5 reanalysis product[32], and hereafter are simply referred to as ERA5. In this analysis, the pace of the shifting seasons in lakes is also compared with those derived from surface air temperature from ERA5. During the satellite data-taking period, fine-scale spatially resolved satellite-derived lake surface temperatures from the Great Lakes of North America is also analysed (1995–2021). To investigate future (2021–2099) changes in the pace of the shifting seasons, modelled lake surface temperatures from the Inter-Sectoral Impact Model Intercomparison Project (ISIMIP) phase 2b Lake Sector is analysed. These data include surface water temperature simulations from ~17,000 representative lakes worldwide as well as from 50 lakes with detailed bathymetry and observational validation data[33]. The ISIMIP projections are based on an ensemble of lake models, each forced with climate data from an ensemble of 20th and 21st century climate projections, under different anthropogenic greenhouse gas emission scenarios (Representative Concentration Pathway, RCP): RCP 2.6 (low-emission scenario), 6.0 (medium-high-emission), and 8.5 (high-emission), as well as a climate influenced solely by natural processes with no anthropogenic influence, defined according to a pre-industrial control simulation.

## Results

### Shifting seasonality of lakes during the historic period

To begin this investigation, historic seasonal shifts in spring and autumn lake surface temperatures using the ERA5 data (1980–2021) are calculated. Note that in this analysis I exclude lakes that, during the season of interest, are either (i) ice covered, or (ii) experience a minimal (<0.5 °C month$^{-1}$) climatological seasonal rate of change in surface water temperature (Figs. S2–S5). The latter suggests a relatively marginal seasonal variation in lake surface temperature and thus are excluded in this study[5]. Across the studied sites, the data suggests that spring temperatures have arrived earlier by 2.0 (1.4, 2.8) days decade$^{-1}$ in the Northern Hemisphere (number of representative lakes [$n$] = 33,122) and by 1.9 (0.7, 3.4) days decade$^{-1}$ in the Southern Hemisphere ($n$ = 21,728) from 1980 to 2021 (Fig. 1; Table S1). The summary statistics quoted here represent the medians and interquartile ranges (25th and 75th) of all lakes in the study domain, which were chosen instead of means and standard deviations given the skewness of the distributions (i.e., often with a long tail, representing lakes that experience particularly dramatic seasonal shifts which could skew the results). Note that the number of representative lakes where these metrics could be calculated differ between the Northern and Southern Hemisphere, as well as across seasons, due to the presence of lake ice cover (e.g., lakes in polar regions) and the number of lakes with relatively minimal temperature change during the season of interest (e.g., tropical lakes). Moreover, given these constraints, all the ~150,000 representative lakes are not included in the summary statistics. In turn, it is important to consider the uncertainty associated with each of the statistics quoted, regarding a variable sample size. The calculated shifts in spring lake surface temperatures are comparable to those calculated in local air temperature (i.e., the longitude-latitude grid in which each lake is situated). It is estimated that spring air temperature has arrived earlier by 2.5 (1.3, 4.0) days decade$^{-1}$ and 3.6 (1.0, 8.1) days decade$^{-1}$ in the Northern and Southern Hemisphere, respectively (Table S1). Also apparent in the global scale assessment was a change to the arrival of autumn lake surface temperatures during the study period, at a rate of −1.5 (−2.5, −0.8) days decade$^{-1}$ in the Northern Hemisphere ($n$ = 44,547) and by −1.1 (−1.9, −0.5) days decade$^{-1}$ in the Southern Hemisphere ($n$ = 20,423) (Fig. 1; Table S1) −Note that positive and negative values indicate when the timing of a season has advanced or been delayed, respectively. These estimated shifts in autumn lake surface temperatures are also comparable to calculated shifts in autumn air

temperature. This study suggests that the arrival of autumn air temperature has changed by −1.8 (−2.9, −1.0) days decade$^{-1}$ in the Northern Hemisphere and by −1.6 (−3.2, 0.5) days decade$^{-1}$ in the Southern Hemisphere (Table S1). Intuitively, regions that experience rapid seasonal shifts in lake surface temperature align with those that experience a relatively small seasonal rate of change and a substantial warming of monthly surface water temperature during the season of interest (Figs. S6–S8). For example, lakes in the eastern United States and in western Europe experienced large shifts in the timing of autumn during the study period, driven by rapid lake warming in October and a below average seasonal rate of change at this time of year (Figs. S6–S7). The seasonal rate of change is influenced by (i) the seasonality of surface heat exchange between the lake and the atmosphere (e.g., due to air and lake temperature variations), and (ii) lake depth. Typically, deeper lakes take longer to heat in spring and to cool in autumn, due to their larger thermal inertia[34–37]. In turn, the seasonal rate of change in lake surface temperature is typically slower in these systems.

Whilst the advance and delay in spring and autumn lake surface temperatures, respectively, are substantial, this analysis suggests that seasonal shifts at other times of the year may be even greater. Specifically, dramatic shifts are calculated in the arrival of summer surface water temperatures across the studied lakes from 1980 to 2021 (Fig. 2a, b; Fig. S8; Table S2). Summer lake temperatures arrived earlier by 4.4 (2.4, 7.0) days decade$^{-1}$ in the Northern Hemisphere ($n$ = 63,809) and by 6.4 (3.1, 10.0) days decade$^{-1}$ in the Southern Hemisphere ($n$ = 7453) (Table S2; Fig. S8). Interestingly, these rates of change in lake surface temperature are approximately half those estimated in local air temperature. Notably, this study suggests that summer air temperature has advanced by 9.0 (−0.7, 17.5) days decade$^{-1}$ in the Northern Hemisphere and 12.1 (3.8, 24.2) days decade$^{-1}$ in the Southern Hemisphere (Table S2). However, it is also important to recognize that a lag exists between air and water temperature, particularly for large and deep lakes[36]. This lag can allow for a partial decoupling of air and water temperature at seasonal timescales with, for example, surface water temperatures in large lakes being cooler than the overlying air in summer and warmer in autumn. Given the global-scale advance in the arrival of summer temperatures and the delay in the arrival of autumn temperatures, this analysis suggests a rapid lengthening of the summer season in lakes worldwide (Fig. 2c, d; Table S3). Indeed, this analysis suggests that summer has lengthened by 5.6 (3.6, 8.9) days decade$^{-1}$ in the Northern Hemisphere ($n$ = 34,515) and by 7.6 (4.3, 10.8) days decade$^{-1}$ in the Southern Hemisphere ($n$ = 7404) (Table S3). These rates of change, however, are considerably lower than those estimated in local air temperature (Table S3). Also note that for any given lake, the summer season will lengthen at a rate equal to the sum of the advance in the arrival of summer and the delay in the arrival of autumn. However, as the number of lakes used in the hemispheric scale calculations above differ (e.g., 63,809 in Northern Hemisphere summer and 44,547 Northern Hemisphere autumn), a change in the length of summer and an advance/delay in the start of summer/autumn will not be directly comparable.

The ERA5 data used in this global-scale analysis of historic lake surface water temperature change is based on simulations from a one-dimensional lake model that was developed primarily to simulate the thermal dynamics of small and relatively shallow lakes (see Methods). In turn, some of the largest lakes of the world are not included in the global-scale historic simulations and thus not represented in the summary statistics quoted above. To investigate changes in some of the world's largest lakes, satellite-derived lake surface water temperature observations are analysed. Notably, to complement the global analysis, I now look in closer detail at the largest group of freshwater lakes on Earth, The Great Lakes of North America. Using spatially resolved (~1.8 km resolution) and uninterrupted daily satellite-derived lake surface water temperature time series, acquired from the Great Lakes Surface Environmental Analysis[38], I calculate

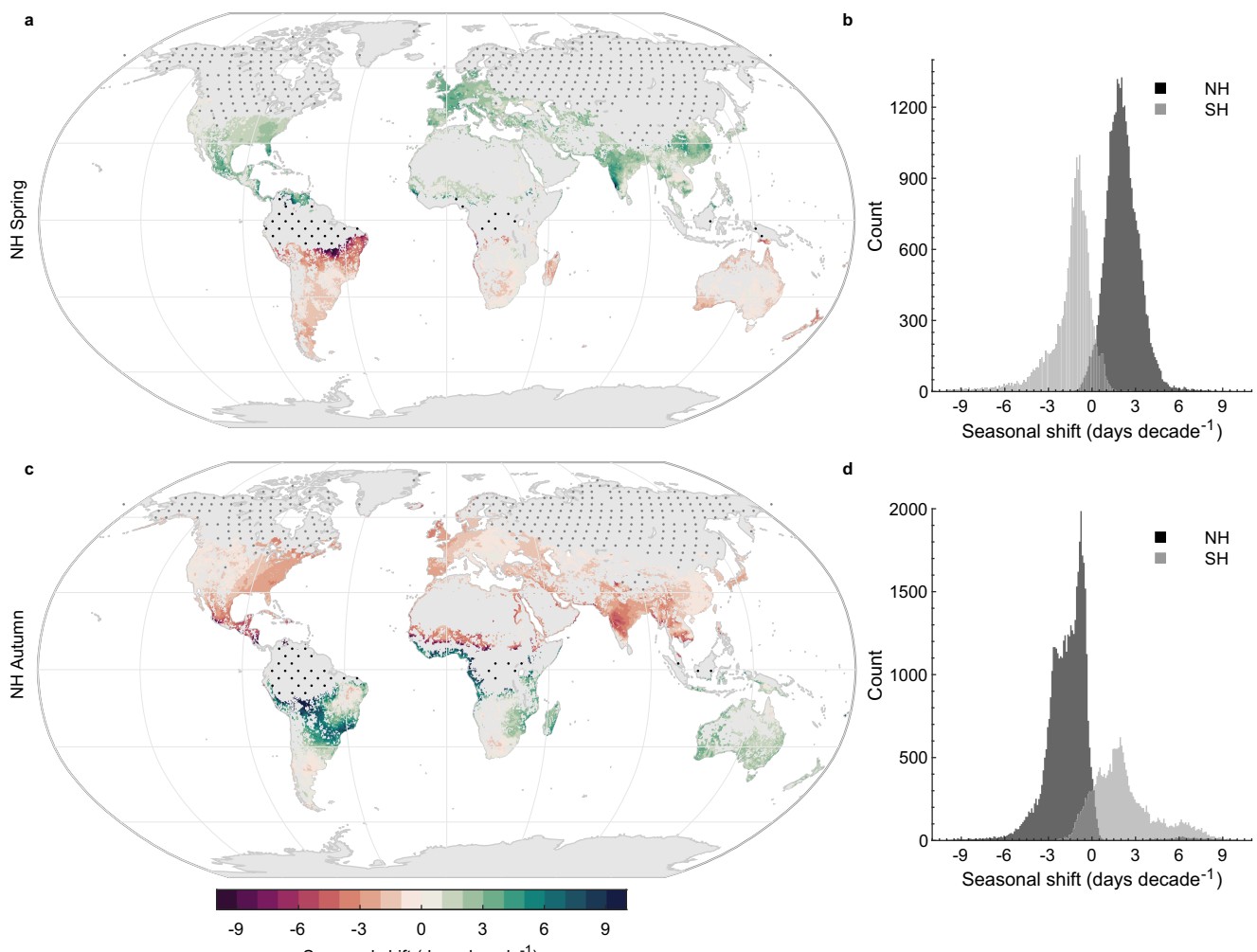

**Fig. 1 | Seasonal shifts in the timing of spring and autumn in lakes.** Shown are the seasonal shifts (days decade⁻¹) in the timing of spring (March–May in Northern Hemisphere [NH] and September-November in the Southern Hemisphere [SH]), and autumn (September–November in the Northern Hemisphere and March–May in Southern Hemisphere) temperatures in lakes worldwide during the historic to contemporary period (1980–2021). Panels (**a**) and (**b**) show the seasonal shifts in NH spring and SH autumn, and panels (**c**) and (**d**) show the seasonal shifts in NH autumn and SH spring. Positive and negative values indicate when the timing of a season has advanced or been delayed, respectively. Stipple markings in panels (**a**) and (**c**) represents regions that were not included in the analysis, either due to the presence of lake ice cover (grey points) or when lakes experienced a minimal (<0.5 °C month⁻¹) seasonal rate of change in surface water temperature (black points) during the season of interest (see Figs. S2–S8). Histograms in panels (**b**) and (**d**) demonstrate calculated shifts in both NH and SH lakes.

rapid seasonal shifts from 1995 to 2021 (Fig. 3). Although the satellite data cover a shorter time period and thus are not directly comparable to the global-scale assessment discussed above, these are essential to quantify observed changes in the timing of seasonal temperatures in large lakes. Note that as the Great Lakes often experience ice cover for several months of the year, typically lasting well into spring, only changes to the timing of summer and autumn (following the same definitions as previously described for lakes worldwide) are investigated here. Also, similar to the global-scale analysis, only regions within the Great Lakes that are ice free and experience a climatological seasonal rate of change of >0.5 °C month⁻¹ during the season of interest is considered (Figs. S9–S12). In the Great Lakes, summer lake surface temperatures arrived earlier by 3.4 (2.5, 5.1) days decade⁻¹ in Lake Superior, 4.2 (3.1, 5.3) days decade⁻¹ in Lake Michigan, 5.0 (3.9, 6.5) days decade⁻¹ in Lake Huron, 5.5 (4.8, 6.5) days decade⁻¹ in Lake Erie, and 6.8 (5.9, 8.5) days decade⁻¹ in Lake Ontario between 1995 and 2021. Dramatic changes were also observed in autumn, with the arrival of autumnal lake surface temperatures changing by −5.1 (−6.0, −3.9) days decade⁻¹ in Lake Superior, −3.5 (−5.1, −2.5) days decade⁻¹ in Lake Michigan, −4.0 (−5.4, −2.8) days decade⁻¹ in Lake Huron, −3.5 (−3.8, −3.1) days decade⁻¹ in Lake Erie, and −4.2 (−5.0, −3.6) days decade⁻¹ in

Lake Ontario. As previously discussed, the pace of the shifting seasons is influenced by the seasonal rate of change and the monthly temperature trend (Figs. S13–15). Similar to the global-scale analyses, a lengthening of the summer season was also calculated in each of the Great Lakes (Table S4). Since 1995, the period of summer temperatures has lengthened most dramatically in Lake Ontario, at a rate of 11.0 (9.6, 13.2) days decade⁻¹. Each of the other Great Lakes experienced a lengthening of the season with summer temperatures at a rate of approximately 8 days decade⁻¹.

The calculations above for the Great Lakes focused on the same definition of the seasons as the global-scale analysis. However, it is important to consider that whilst maximum lake surface water temperature occurs typically during Jun–Aug in many Northern Hemisphere lakes[39], some large and deep lakes can experience a thermal maximum (and thus arguably when "summer" occurs) much later in the year. For example, in the Great Lakes, maximum surface water temperature can occur in September[40,41], thus outside of the summer season defined previously. This is related to the fact that phenology in lakes is strongly influenced by stratification, not solely air temperature, and the vast thermal inertia of large water bodies can lead to maximum surface temperatures occurring later in the year. To consider this

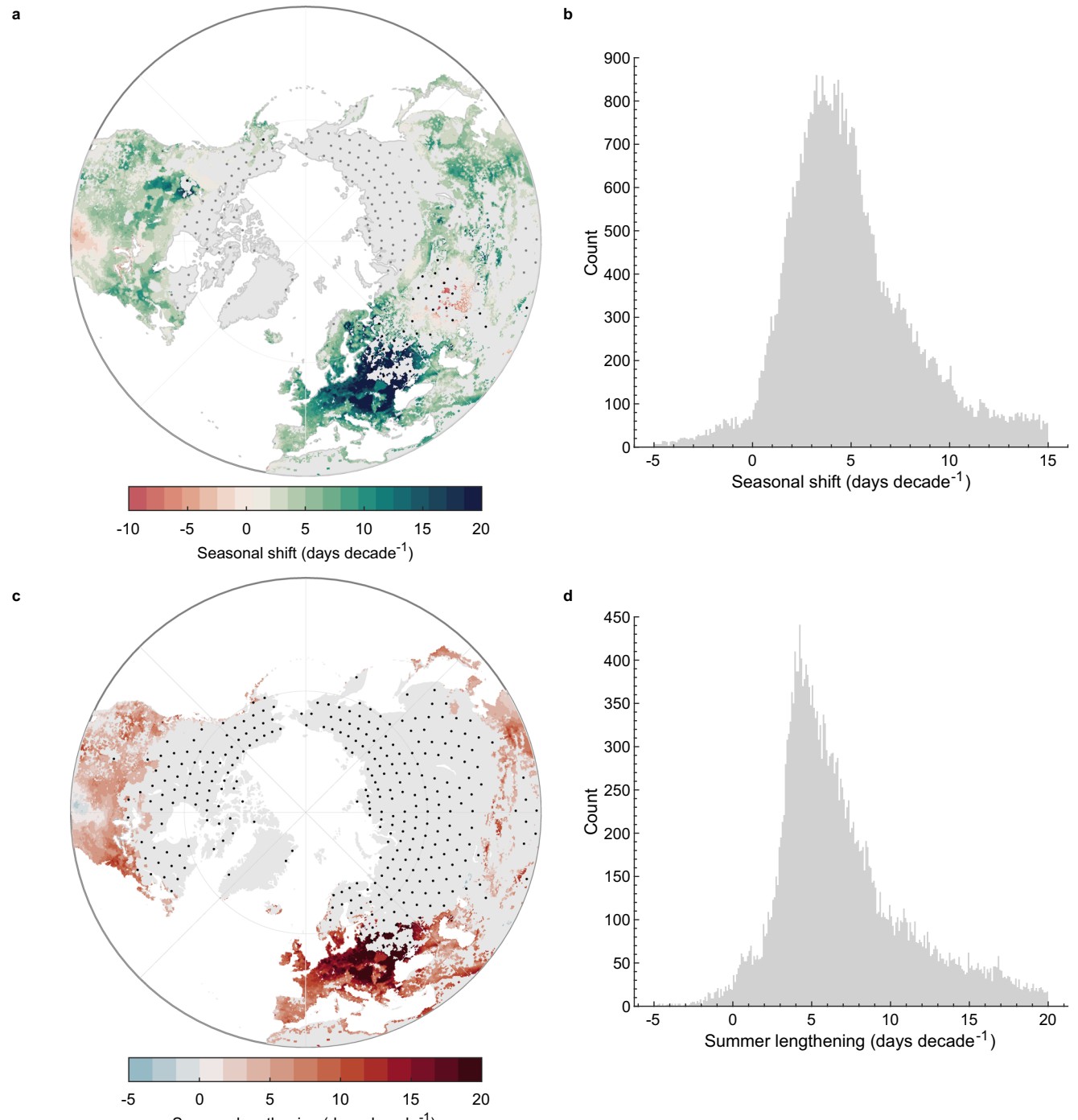

**Fig. 2 | Alterations to the timing and duration of summer in lakes.** Shown are (**a**, **b**) calculated shifts (days decade⁻¹) in the timing of summer (June–August) temperatures across Northern Hemisphere (NH) lakes during the historic to contemporary period (1980–2021). Positive and negative values indicate when the timing of summer has advanced or been delayed, respectively. Results for Southern Hemisphere (SH) lakes are shown in Fig. S8. Also shown in panels (**c**) and (**d**) are calculated changes to the duration of summer, estimated as the difference between the seasonal shifts in the start of summer and the start of autumn (September–November). Positive and negative values indicate when the summer season has lengthened or shortened, respectively. Stipple markings represents regions that were not included in the analysis, either due to the presence of lake ice cover (grey points) or when lakes experienced a minimal (<0.5 °C month⁻¹) seasonal rate of change in surface water temperature (black points) during the season of interest.

feature of the seasonal temperature cycle of the Great Lakes, I now repeat the analysis above of changes in the timing of seasonal temperatures but considering a variable definition of the seasons. Notably, to ensure that the seasonality of lake surface temperature is captured across the Great Lakes, I repeat the analysis but now consider a moving window of months, whereby a 3-month wide window is used but spanning through the entire year. For example, summer could be

defined as Jun–Aug or Jul–Sep, the latter being particularly appropriate for Lake Superior. Following the approach described previously, here I only consider regions within the Great Lakes that are ice free and experience a seasonal rate of change of >0.5 °C month⁻¹ during the season of interest (Figs. S16–S20). Summary statistics for each of the three-month periods throughout the year is shown in Table S5 and Fig. S21. Overall, the patterns of change between 1995 and 2021, typically

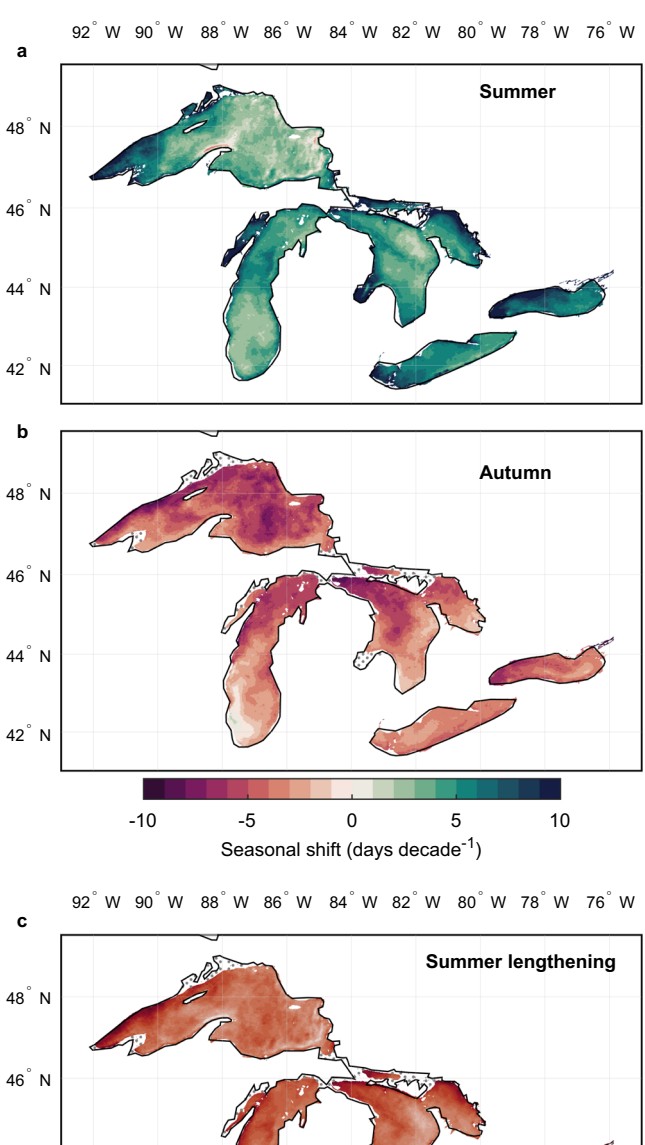

**Fig. 3 | The pace of shifting seasons in the North American Great Lakes.** Shown are the seasonal shifts (days decade⁻¹) in the timing of (**a**) summer (June–August) and (**b**) autumn (September–November) temperatures during the satellite data-taking period (1995–2021). Positive and negative values indicate when the timing of a season has advanced or been delayed, respectively. Also shown in panel **c** are calculated changes to the duration of summer, which was estimated as the difference between seasonal shifts in the start of summer and of autumn (as shown in panels (**a**) and (**b**)). Positive and negative values indicate when the summer season has lengthened or shortened, respectively. Stipple markings represents regions that were not included in the analysis, either due to the presence of lake ice cover (grey points) or when lakes experienced a minimal (<0.5 °C month⁻¹) seasonal rate of change in surface water temperature (black points) during the season of interest (see Figs. S9–S15).

follow the same direction as the previously defined seasons, but with differences in the calculated rates of change. For example, if one was to define the summer season in Lake Superior as Jul-Sep, summer temperatures would have arrived earlier by 5.8 days decade⁻¹ (Table S5).

Similarly, if autumn was defined as Sep–Nov, autumn temperatures would have arrived later by 5.1 days decade⁻¹ (Table S5). However, a complication that arises when the seasons are defined according to these other months in the Great Lakes, is that the percentage of valid pixels is reduced (often considerably) either due to the presence of ice cover or a minimal climatological seasonal rate of change, which can inevitably bias the summary statistics. It is only when summer is defined as Jun–Aug, that 100% lake coverage is considered for the analysis (Table S5). Furthermore, this variable season approach clearly demonstrates differences in the seasonality of surface water temperature in the Great Lakes with, for example, Lakes Superior and Erie following different trajectories (seasonal warming/cooling) during Jul-Sep, largely reflecting differences in their thermal mass—i.e., Lake Superior is still warming at this time of year whereas Erie has started cooling. This feature also illustrates the complexity of analysing seasonal shifts in lake temperature using a fixed season definition and should be considered when interpreting some of the main findings of this study. For completeness, I now repeat the global-scale historical analysis of seasonal temperature changes (i.e., using data from ERA5) but applying the 3-month wide window for defining the seasons— similar to that described above for the Great Lakes. These results are presented in Figs. S22–24 and Table S6.

**Shifting seasons under future climate change**
Having demonstrated the influence of historical to contemporary (1980–2021) climate change on the arrival of the seasons in lake environments, I now investigate projected changes under future climatic forcing (RCPs 2.6, 6.0 and 8.5) from 2021 to 2099. Ultimately, here I investigate by how much more will the seasons likely change under continued warming this century, and how different greenhouse gas emission scenarios can influence the magnitude of projected change. Moreover, I compare the future simulations under the different RCPs with those simulated with pre-industrial climatic forcing, that is a counterfactual world where climatic variations are influenced solely by natural variability and not exposed to anthropogenic greenhouse warming. Similar to the global-scale simulations investigated during the historic period, the future global-scale simulations represent an aggregated 'typical lake' for each model grid cell, simulating the average lake thermal environment in that location using the grid cell's climate forcing and a representative lake morphometry (see Methods). Note that, in this section, when the arrival or length of a specific season is discussed, these specifically refer to the arrival/ length of current seasonal temperatures. The future simulations suggest that current spring temperatures in Northern Hemisphere lakes will arrive earlier by 0.3 (0.1, 0.5), 1.8 (1.6, 2.4), and 3.3 (2.8, 4.0) days decade⁻¹ this century under RCP 2.6, 6.0 and 8.5, respectively (Fig. 4a, e; Table S7). The latter is -2 magnitudes greater than that projected within a pre-industrial climate at −0.04 (−0.2, 0.2) days decade⁻¹. Maps of the projected changes under the different climate scenarios are shown in Figs. S25–28. The rate of change in summer lake temperatures across the Northern Hemisphere are projected to be even greater, with current summer temperatures arriving earlier by 0.5 (0.3, 0.8), 4.8 (3.2, 6.1), and 8.3 (5.9, 10.2) days decade⁻¹ under RCP 2.6, 6.0 and 8.5, respectively (Fig. 4b, f) which, again, are an order of magnitude greater than would occur in a naturally varying climate (Table S8). These projections suggest similar differences between the factual (i.e., influenced by both natural and anthropogenic forcing) and counterfactual worlds in autumn lake temperatures, with the timing of current autumn temperatures changing by 0.02 (−0.1, 0.1), −0.2 (−0.3, −0.1), −1.7 (−1.9, −1.5), and −3.1 (−3.4, −2.6) days decade⁻¹ under pre-industrial and RCP 2.6, 6.0 and 8.5, respectively (Fig. 4c, g; Table S7). Regarding the length of periods with current summer temperatures, the future projections suggest a lengthening of 0.8 (0.4, 1.1), 7.2 (6.0, 8.3), and 12.1 (10.3, 13.8) days decade⁻¹ under RCP 2.6, 6.0 and 8.5, respectively across Northern Hemisphere lakes (Fig. 4d, h; Table S9; Fig. S29).

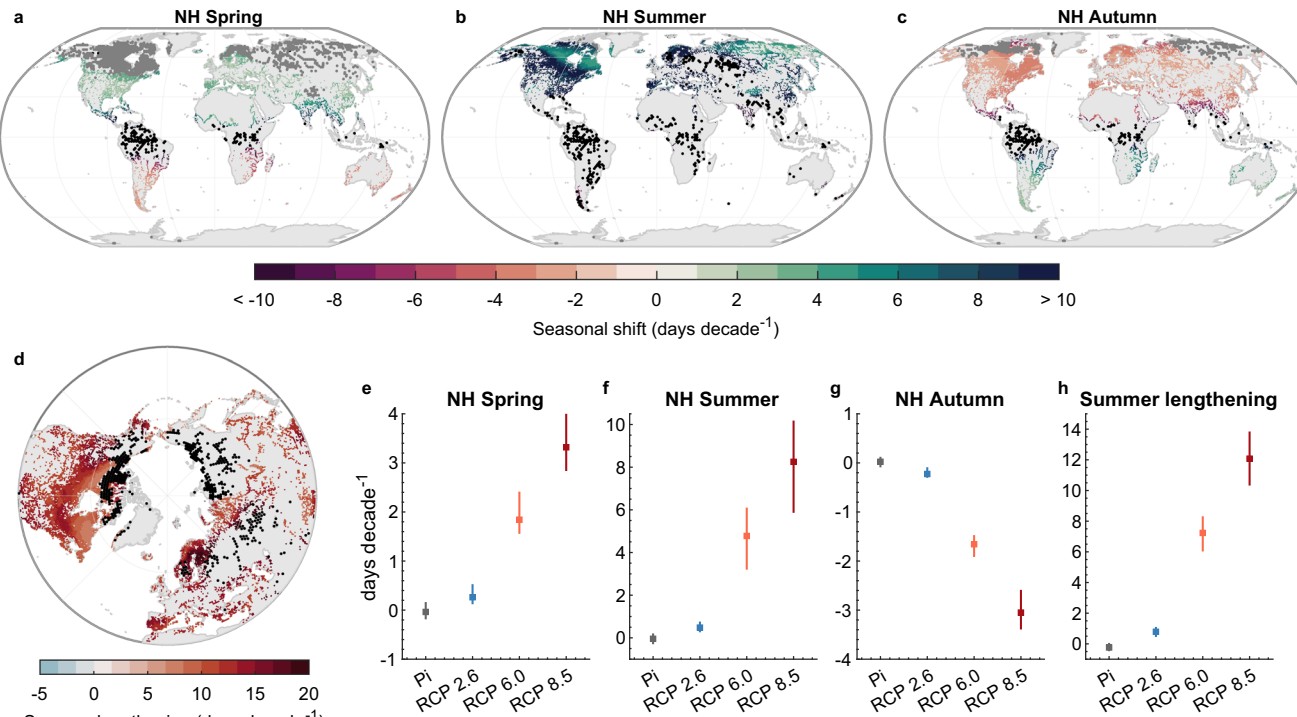

**Fig. 4 | The pace of shifting seasons in lakes under future climate change.** Shown are future (2021–2099) seasonal shifts (days decade$^{-1}$) in the timing of (**a**) spring (March–May in Northern Hemisphere [NH] and September–November in the Southern Hemisphere [SH]), **b** summer (June–August in the NH and December–February in SH), and (**c**) autumn (September–November in the NH and March–May in SH) temperatures in lakes under Representative Concentration Pathway (RCP) 8.5. Positive and negative values indicate when the timing of a season will advance or be delayed, respectively. Also shown in panel **d** are projected changes to the duration of summer under RCP 8.5, which was estimated as the difference between the seasonal shift in the start of summer (shown in (**b**)) and the autumn (shown in (**c**)), in NH lakes. Positive and negative values indicate when the summer season is projected to lengthen or shorten, respectively. Stipple markings represents regions that were not included in the analysis, either due to the presence of lake ice cover (grey points) or when lakes experienced a minimal (<0.5 °C month$^{-1}$) seasonal rate of change in surface water temperature (black points) during the season of interest. In panels **e–h** I show summary statistics (medians and interquartile ranges) for the future simulations under all RCPs (2.6, 6.0 and 8.5) as well as the pre-industrial control simulation.

Similar changes to seasonal temperatures are also projected for Southern Hemisphere lakes, which are shown in the Supplementary Information (Figs. S25–28). Note that in some instances, the estimated pace of the shifting seasons is faster during the historic (1980–2021) than the future (2021–2099) period, notably under RCP 2.6. This is largely due to the specific features of the RCP scenario. Specifically, RCP 2.6 is a low-emission scenario where emissions start declining at around 2020, and hence the rate of warming is reduced thereafter. RCP 6.0 is a medium-high-emission scenario where emissions peak at around 2080 and then decline, and RCP 8.5 is a high-emission scenario where emissions continue to rise throughout the 21st century. Thus, when calculating the shifting seasons from 2021 to 2099 using RCP 2.6 the rate of change in monthly (e.g., April) water temperature will be less than during the historic period (1980–2021) when lakes are experiencing accelerated warming throughout. Similarly, for RCP 6.0 the decrease in greenhouse gas emissions, and thus global warming, toward the year 2080 will influence the calculated rate of change in lake temperature.

The lake surface temperature simulations analysed above were used to investigate projected changes in the timing of seasonal temperatures at large spatial scales by describing how a representative lake in a particular location will likely respond to future climate change. These gridded simulations are essential to quantify lake responses to climate change across large spatial scales[42–47]. This information is also critical for enabling comparisons with projected temperature changes across other surfaces of the Earth, including sea surface temperature and surface air temperature over land, both of which are often reported as gridded averages in climate impact studies[5,48,49]. However, for local-scale assessments, water managers

often require information that are specific to an individual lake, which may differ to the representative lake of that region. To investigate projected changes in individual lakes, I now analyse surface temperature simulations from 50 lakes where detailed lake-specific projections were available from the ISIMIP2b Lake Sector (see Methods). The studied sites are among some of the best-monitored lakes in the world including, among others, Lake Tahoe (California/Nevada, USA), Lake Mendota (Wisconsin, USA), Lake Rotorua (Bay of Plenty Region, New Zealand), and Windermere (English Lake District, UK). In Tables S10–S13 the projected seasonal shifts in each studied lake under the different RCP scenarios are shown. Among the 50 studied lakes, the multi-model simulations project rapid changes, often at rates much higher than those reported at the hemispheric scale. For example, in Lake Tahoe, a large (surface area of 495 km$^2$) and deep (maximum depth of 501 m) sub-alpine lake situated on the border of California/ Nevada (USA), the timing of current spring temperatures is projected to advance by 9.8 days decade$^{-1}$ from 2021 to 2099 under RCP 8.5. This is an order of magnitude greater than projected within a pre-industrial climate (Table S10). Moreover, the timing of current summer lake surface temperatures is projected to change by 5.9 days decade$^{-1}$ and the timing of current autumn temperatures to change by −7.4 days decade$^{-1}$ under RCP 8.5. In turn, the length of periods with current summer temperatures is projected to lengthen by approximately two weeks (13.3 days) per decade this century. Other well-studied lakes from around the world are projected to experience similar long-term changes (Tables S10–S13). Indeed, all the studied lakes are projected to experience an advance in the timing of current spring and summer temperatures and a delay in current autumn temperatures this century under RCP 8.5 (Tables S10–S13). However, these simulations suggest

that the magnitude of projected change will be considerably less under RCP 2.6, compared to the other emission scenarios (Tables S10–S13). Specifically, across the studied lakes, the average arrival of current spring and autumn temperatures (advance and delay, respectively) is 20 times lower under RCP 2.6 than 8.5. Moreover, the arrival of current summer temperatures, as well as the length of the summer season, is 50 times lower under RCP 2.6 than 8.5.

## Discussion

A shift in the timing of seasonal temperatures, as has been described in this study, will likely have numerous implications for lake ecosystems. For example, an earlier arrival of current spring temperatures would likely lead to an earlier onset of thermal stratification in monomictic or dimictic lakes (i.e., those that experience one or two mixing events per year, respectively)[41,50,51], enhanced nutrient release from the sediments into the water column[52], alterations to the dynamics of phytoplankton and zooplankton[53–56], and potentially favouring the increased occurrence of cyanobacteria[57,58]. Changes in algal phenology in spring may also result in a decoupling of trophic relationships throughout the entire year[28,29]. A lengthening of the period with current summer temperatures would be expected to have both negative and positive effects on lake ecosystems, depending on the specific species. For example, a prolonging of current summer temperatures would be expected to benefit ectotherms by prolonging the growing season, particularly at high latitudes where it is historically short[59,60]. Specifically, an advance in the timing of current spring temperatures and a delay in current autumn temperatures, and thus a lengthening of summer, is likely to have positive effects on ectotherms in northern regions by increasing fitness via several pathways[61–63], as well as increase the spatial extent and duration of preferred thermal habitat[64]. However, as noted above, this could also lead to phenological mismatches in critical activities, with widespread ramifications across the food web[24,28,29,65,66]. A delay in the arrival of current autumn temperatures could influence lake evaporation rates[67–69], lead to a delay in the timing of autumnal mixing[15,70,71] and in the formation of lake ice cover[72]. Moreover, warmer autumn temperatures in eutrophic lakes could lead to algal blooms occurring much later in the year[73]. It is important to note, however, that the eco-physiological responses expected from these simulated alterations in seasonal temperature will likely differ across lakes (e.g., eutrophic vs oligotrophic systems and deep vs shallow), and will also depend on attributes such as species adaptive capacity, the ability of aquatic species to migrate[74,75], and the disruption of ecological interactions[76]. However, when considering these ecological responses of the studied lakes to shifts in the timing of seasonal temperatures, it is also important to consider that there may be some bias in the results given the definition of the averaging window. This can have an important impact on quantifying the shifts in the timing of seasonal temperatures and, hence, the ecological and biogeochemical implications.

To maintain the existence of resilient and productive lake ecosystems and to prevent many lake regions being adversely affected by the projected changes in seasonal temperatures reported here, particularly when combined with the host of other natural and anthropogenic effects on lakes, such as eutrophication, acidification, overfishing, alterations in thermal stratification and vertical mixing, and changes in water availability including lake size[77], global warming needs to be severely limited. Indeed, this analysis demonstrated that the projected changes in the seasons will be far less dramatic under a low-emission scenario. This research adds a new dimension to this topic by illustrating the shifting seasons in lakes. Ultimately, this analysis of changes to the physical environment of lakes points towards emerging challenges to lake biodiversity, which may be threatened by the accumulated negative effects of rapid seasonal shifts, especially where such conditions coincide with species-rich regions. In contrast, regions with slower seasonal shifts, may be important repositories for biodiversity this century.

## Methods

### Historic to contemporary lake surface water temperature

Lake surface water temperatures from 1980 to 2021 were analysed from ERA5, which were available at a 0.25° by 0.25° longitude-latitude grid resolution[32]. Lake surface temperatures were simulated in ERA5 via the Freshwater Lake model, FLake[78,79], which is implemented within the Hydrology Tiled ECMWF Scheme for Surface Exchanges over Land[80] of the ECMWF Integrated Forecasting System (IFS). Specifically, lakes simulated within each 0.25° grid are based on the mean depth and surface area of all known lakes in that region. These simulations therefore represent an aggregated 'typical lake' for each 0.25° grid, simulating the average lake thermal environment in that location using the grid cell's climate forcing, i.e., by coupling FLake to the grid cell's atmospheric module. Meteorological forcing required for FLake includes air temperature, humidity, wind speed, shortwave radiation, and downward longwave radiation. The lake grid cells investigated in this study include those with a >0% lake cover fraction in the IFS and those where the land-sea mask of ERA5 identified <50% of the grid as land. Moreover, as FLake was developed to simulate the thermal dynamics of lakes shallower than ~60 m (see for example ref. 80.), any deeper representative lakes were removed from the analysis. In total, 155,333 representative lakes simulated via ERA5 are investigated in this study. ERA5 global lake temperatures have been extensively validated in previous studies[43,44,81]. In this study I also investigate air temperature data from ERA5, which are used to compare with the lake surface temperature simulations.

### Lake surface water temperature during the satellite data taking period

Spatially resolved daily lake surface water temperatures from the North American Great Lakes (situated in the upper mid-east region of North America) were acquired from the Great Lakes surface environmental analysis (GLSEA) for the period 1995–2021[38]. This data set includes lake surface temperature observations for Lakes Superior (average depth = 149 m, surface area = 82,103 km$^2$), Ontario (average depth = 86 m, surface area = 18,960 km$^2$), Michigan (average depth = 85 m, surface area = 58,030 km$^2$), Huron (average depth = 59 m, surface area = 59,600 km$^2$), and Erie (average depth = 19 m, surface area = 25,744 km$^2$). This data is commonly used for investigating long-term surface temperature trends in the Great Lakes. The GLSEA is a ~1.8 km resolution satellite-derived surface temperature product that provides uninterrupted time series of daily lake surface temperatures. The satellite data set is based on the NOAA advanced very high-resolution radiometer (AVHRR), a sensor aboard NOAA's Polar Orbiting Environmental Satellites.

### Future simulations of lake surface water temperature

To investigate future (2021–2099) changes in the pace of the shifting seasons, modelled lake surface temperatures from the Inter-Sectoral Impact Model Intercomparison Project (ISIMIP) phase 2b Lake Sector are analysed. These include surface water temperature simulations for 17,436 representative lakes worldwide as well as from 50 lakes with detailed bathymetry and observational validation data. An in-depth explanation of the ISIMIP Lake Sector is given by ref. 33. All ISIMIP Lake Sector projections investigated in this study were simulated via a lake model (see below) driven by an ensemble of bias-corrected climate projections, namely GFDL-ESM2M, HadGEM2-ES, IPSL-CM5A-LR, and MIROC5. Future projections were available under three greenhouse gas emission scenarios: Representative Concentration Pathway Representative Concentration Pathway (RCP 2.6) (low-emission scenario), 6.0 (medium-high-emission), and 8.5 (high-emission). These pathways encompass a range of potential future global radiative forcing from anthropogenic greenhouse gases and aerosols, and results span a range of potential impacts on lake temperature. In addition, the ISIMIP2b simulations include climate projections that are influenced

solely by natural processes with no anthropogenic influence, defined according to a pre-industrial control simulation. The data used to drive the lake models in ISIMIP2b included projections of air temperature at 2 m, wind speed at 10 m, surface solar and thermal radiation, and specific humidity, which were available at a daily resolution. The 17,436 representative lakes investigated in this study were simulated at a 0.5°-by-0.5° grid resolution (i.e., the spatial resolution of the climate projections) with the SimStrat-UoG model. The dataset used to describe the size distribution of all lakes within each 0.5° grid has a horizontal resolution of 30 arc seconds, and include all known lakes equal or greater than this size threshold. Lake-specific simulations from 50 lakes which had detailed bathymetry and validation data were also analysed. Lake surface temperatures for these lakes were simulated by a suite of independently developed lake models: (i) FLake, (ii) General Lake Model (GLM), (iii) General Ocean Turbulence Model (GOTM), and (iv) SimStrat. For both sets of simulations, the pace of the shifting seasons (see below) was calculated independently for each lake-climate model combination and then averaged across the lake-climate model ensemble.

### Calculating the pace of shifting seasons

In this study, I present seasonal shifts in the timing of spring (March–May in Northern Hemisphere and September–November in the Southern Hemisphere), summer (June–August in Northern Hemisphere and December–February in the Southern Hemisphere), and autumn (September–November in the Northern Hemisphere and March–May in Southern Hemisphere) in lakes worldwide. The seasonal shift in lake surface water temperature was calculated for the season of interest by dividing the long-term trends in monthly surface water temperature (°C year$^{-1}$) by the climatological seasonal rate of change (°C month$^{-1}$) centred on each month[5]. The climatological seasonal rate of change is given by half the difference in lake temperature between the preceding and following months during the season of interest, averaged throughout the study period. For example, to estimate changes to the timing of spring in Northern Hemisphere lakes, I calculated the long-term trend in lake surface water temperature during April (e.g., from 1980 to 2021) divided by the climatological seasonal rate of change between March and May. Specifically, the latter is a climatology of the seasonal rate of change in water temperature between the months of interest. The same approach was then applied to calculate the seasonal shifts during the other seasons. Resulting values for seasonal shifts (month year$^{-1}$) were converted to days decade$^{-1}$ by multiplying by 10 years, 365.25 days year$^{-1}$ and dividing by 12 months. These calculations were performed in R[82] using the 'shiftTime' function available in the 'VoCC' package[83] and follow the methods described by ref. [5]. Similar to ref. [5], I ignore climatic regions where the seasonal rate of change in temperature is small, which is characteristic of tropical lakes with low seasonal variability in water temperature[39]. Here, I only investigate lakes with a seasonal rate of change of greater than 0.5 °C month$^{-1}$. Lower seasonal variations in temperature imply marginal seasonal variations and thus are not of interest in this study[5]. I also excluded from the analysis lakes that experienced ice cover during the season of interest. To exclude the influence of, for example, one anomalous year I only excluded a lake from the analysis when 90% of all years experienced ice cover during the season. The analysis described above was also repeated for different definitions of the seasons. Specifically, to ensure that the seasonality of lake surface temperature is captured across lakes, I repeat the analysis described above, but considering a moving window of months, whereby a 3-month wide window is used but spanning through the entire year.

### Data availability

ERA5 air and lake surface water temperature data used in this study are available from https://cds.climate.copernicus.eu/cdsapp#!/dataset/ reanalysis-era5-single-levels?tab=overview. ISIMIP2b Lake Sector local simulations are available here https://doi.org/10.48364/ISIMIP.563533. ISIMIP Lake Sector global simulations are available here: https://doi.org/10.48364/ISIMIP.931371. Lake surface temperature data from the Laurentian Great Lakes are available at https://coastwatch.glerl.noaa.gov/glsea/.

### Code availability

The code used to produce the figures in this paper is available from the corresponding author upon request.

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

## Acknowledgements

This work was supported by a UKRI Natural Environment Research Council (NERC) Independent Research Fellowship awarded to RIW [grant number NE/T011246/1]. For their roles in producing, coordinating, and making available the ISIMIP climate scenarios, I acknowledge the support of the ISIMIP cross sectoral science team. For producing the future lake model simulations, I thank the ISIMIP Lake Sector modellers (Simstrat-UoG: Marjorie Perroud; FLake: Georgiy Kirillin, Tom Shatwell; General Lake Model: Robert Ladwig, Tadhg Moore; General Ocean Turbulence Model: Robert Ladwig, Tadhg Moore; SimStrat: C. Love Råman Vinnå), and the ISIMIP Lake sector coordinators (Gosia Golub, Rafael Marcé, Wim Thiery, Don Pierson, Daniel Mercado-Bettin). I also thank other members of the ISIMIP Lake Sector who contributed to the simulations.

## Author contributions

R.I.W. initiated the project, led the data analysis, design of visualizations, and writing of the manuscript.

## Competing interests

The author declares no competing interests.
