## [Peer Review File · Nature Communications]

The pace of shifting seasons in lakesReviewer #1 (Remarks to the Author):

The manuscript presents an assessment of how the lake surface water temperature is changing in key moments of the year by means of shift seasons in lakes. The pace of shifting seasons in lakes has been calculated for the historic period (using ERA5 data) and for the future using simulations under three different RCPs. Moreover, the author presents the same analysis using satellite spatially distributed data for the Laurentian Great Lakes.

The topic of the manuscript is very relevant and interesting, and the global scale of the analysis allows also drawing large-scale conclusions. The topic is in line with the journal scope and the manuscript is well written. I recommend the publication of this study but I think that some aspects should be clarified before publishing the manuscript.

In the following paragraph, I collect the major and the minor issues from my review.

Major issues:

- Definition of seasons.

The analysis takes as bases the definition of seasons as following: spring (March-May in Northern Hemisphere and September-November in the Southern Hemisphere), summer (June-August in Northern Hemisphere and December-February in the Southern Hemisphere), and autumn (September-November in the Northern Hemisphere and March-May in Southern Hemisphere). Generally this is accepted, however, some recent studies show that depth plays an important role in defining when the temperature reaches its maximum for example so when summer occurs for example. This is related to the fact that phenology in lakes, is strongly set by stratification as well, not only temperature as in land ecosystems. To take into account this I would suggest to try to do the same analysis considering a moving window of months, meaning to use still 3 months wide window but spanning through the all year to be sure that the real summer, spring and autumn of each lake is properly captured. Ex: JFM, FMA, MAM, ...

- Description of how to calculate the pace of shifting seasons.

The description of how the pace of shifting season has been calculated (from L366 on) it is an important point for the reproducibility of the study. I would suggest to add a step by step description of the calculation method in the supplementary. My main question in this regard would be the following: let's take spring as an example, April is the central month so you calculated the long term trend for April and then divide this by the difference in temperature between March and May but for each year? Meaning that you end up having one value for each year (1981-2020)? How do you treat then this time series? I suggest adding some information on this point.

- Northern and Southern Hemisphere.

The number of lakes or "aggregated lakes" in the two hemisphere could be very different and somehow influence the statistics of the seasonal shifts. I suggest adding information on this.

- Comparison between historic and future pace of shifting seasons.

The comparison with the shifting seasons in air temperature is very interesting and meaningful but I suggest adding a comparison between the historic and future rates. Particularly, I was curious to understand why the historic values reported for example for spring using the historic period (L95) are higher than values for spring using the first 2 RCPs as future scenarios (L189-190). I suggest adding a discussion of the comparison for all seasons.

Minor issues:

- L11 – "Much of the focus of...have focused...". I suggest changing one of the 2 "focus".
- L11-12 – "mean temperature" means temporal mean in this case right? I suggest to specify because the reader could interpret this as spatial mean otherwise.
- L71 – "aggregated typical lake" becomes much more clear after reading the Supporting Information so I suggest adding a reference to that explanation by directly adding part of the explanation in the main text.
- L168 – "later by -5.1". I suggest changing it to earlier and adding a positive number. This would help the reader.
- L189-190 –
-
- Figure 1 – I suggest adding "spring" on the left of panel a and "autumn" on the left of panel c to

have the figure more self-explicative.

- Figure 4 – I suggest using “summer lengthening” instead of “longer summer” for the colorbar and consistently in the manuscript. Also, the abbreviations NH and SH should be reported in the caption.
- Table S1 – I suggest adding the analogous table with the future scenarios. I know that values for single lakes are reported in tables from S5 on but having an aggregated table for the Northern and Southern Hemispheres would help the comparison with the historic periods.

Reviewer #2 (Remarks to the Author):

Woolway analyzed past (modeled and observed) and future (modeled) lake surface water temperature data to assess the pace of shifting seasons in lakes worldwide. The analysis has been done considering three seasons (spring, summer, and autumn) and different climate change scenarios (RCP 2.6, 6.0, 8.5 and pre-industrial control simulation). The present study follows the line of recent studies published by the Author and dealing with the effects of climate change on lakes' temperature and stratification phenology, e.g.: Woolway and Merchant (2019, <https://doi.org/10.1038/s41561-019-0322-x>), Woolway and Maberly (2020, <https://doi.org/10.1038/s41558-020-0889-7>); Woolway et al. (2021, <https://doi.org/10.1002/lo2.10231>); Woolway et al. (2021, <https://doi.org/10.1038/s41467-021-22657-4>); Woolway et al. (2022, <https://doi.org/10.1093/biosci/biac052>); Woolway et al. (2022, <https://doi.org/10.1029/2021GL097031>).

The methodological approach used by the Author is the one proposed by Burrow et al. (2011) to assess the pace of shifting climate in marine and terrestrial ecosystems. Burrow et al. (2011) proposed two indicators of climate change shift: the velocity of climate change and the seasonal climate shift. The first indicator has been used by the Author in a previous manuscript (Woolway and Maberly, 2020 <https://doi.org/10.1038/s41558-020-0889-7>). The latter has been used in the present study. I recognize that the Author is brilliant in presenting worldwide trends and patterns using original methodologies or applying existing ones, as in this case. However, I also think that the key message of this manuscript is not new compared to that of the previous studies by the Author, but it is the same presented in a different way. Importantly, I noticed that section "Implications of shifting seasons in lakes" is made of several sentences almost entirely copied-pasted from previous manuscripts of the Authors (Woolway et al., 2021, <https://doi.org/10.1038/s41467-021-22657-4>; Woolway et al., 2022, <https://doi.org/10.1093/biosci/biac052>; Woolway et al., 2022, <https://doi.org/10.1029/2021GL097031>), where the key words of these previous studies (e.g., heatwaves, stratification phenology) have been substituted with the one of the present study (i.e., seasonal shift). This reinforces the idea that the message of this study is not entirely new. This is all the more true when we consider that the literature on the topic is quite vast and increasing at a fast pace, e.g., Jane et al (2022, <https://doi.org/10.1111/gcb.16525>), Oleksy and Richardson (2021, <https://doi.org/10.1029/2020GL090959>), Kraemer et al. (2021, <https://doi.org/10.1038/s41558-021-01060-3>), Dokulil et al (2021, <https://doi.org/10.1007/s10584-021-03085-1>), Piccolroaz et al. (2021, <https://doi.org/10.1016/j.ejrh.2021.100780>), Råman Vinnå et al. (2021, <https://doi.org/10.1038/s43247-021-00106-w>), Shatwell et al. (2019, <https://doi.org/10.5194/hess-23-1533-2019>), Niedrist et al. (2018, <https://doi.org/10.1007/s10584-018-2328-6>), Butcher et al. (2015, <https://doi.org/10.1007/s10584-015-1326-1>)

As for the methodology:

- as correctly pointed out by the Author, LSWT seasonality is largely influenced by seasonality of surface heat fluxes, timing of ice-cover and lake depth. I think that the approach used by the Author (FigS1) at the global scale can seriously hide these effects, resulting in a flattening (simplification) of the results in terms of deep understanding/representation of the processes and of the inter-lake variability. As an example, the three seasons investigated by the Author correspond to very different periods of the LSWT annual cycle of the Great Lakes (see e.g., Fig 1 in Cannon et al., 2019, <https://doi.org/10.1029/2019GL082916>). This certainly adds complexity to the processes and to the analysis.

- Fig1 b and d: showing NH and SH with different colors would be useful to appreciate to what extent the bimodal distribution is explained by latitude above/below the equator.

Minor comments:

- L11: "focus", "focused"

- L13: "event" to "even"

- FigS1: this is exactly as in Burrow et al. (2011), except for minor details. I would not include June and write that the schematic applies for spring.

- FigS6, L690 remove "are"

Reviewer #3 (Remarks to the Author):

The submitted manuscript reports a quality work carried out with the aim of filling the knowledge gap about present and future climatic shifts in lakes. The work is of great topicality and significance to the field. The manuscript is original, very well organized, well written and clear.

Major Issue

I have a single fundamental question about the work and it was because of this issue that I took longer than I intended to review the article. My apologies to the authors and editors. The question is about the definition of shifting seasons and how to quantify it. I followed the references and understood that it is a methodology already widely used in studies about climatic shifts in both marine and terrestrial ecosystems. I recognize that the measure of shifts in temperature, as done in the paper, can be a good quantifier of climate change, but it seems to me a bit abusive to use it as a synonym for "seasonal shift". The concept of season in climate is much more than the recorded temperature. To be more precise, in my opinion what this parameter measures is the shift of the arrival of the temperature of current season. This measure can be seen also as an indicator of the increase in temperature of the season, and not necessarily of a shift of the season. This problem is likely to be magnified in summer as, namely at high latitudes, in the Northern Hemisphere July is the hottest month so the differences between August and June temperatures are small. This question came to me sharply when I analysed the results of the study. For example, when presenting the estimate that the summer season will lengthen at a rate of 12.1-days decade⁻¹ (line 24), means that it is estimated that the duration of the summer will increase by about 100 days at the end of the century, that is, it will last for more than two seasons. In fact, what the study actually indicates is that, at the end of the century, there will be another 100 days with current summer temperatures. This conclusion is, by itself, very interesting and important, but it doesn't seem to me to be the same as saying that summer is twice as long.

I admit that it can be a preciousness, but I would like to suggest that authors pay attention to this issue and mitigate the use of the terms like "season shift", "arrival of spring", "length of Summer". Instead, it seems more appropriate to use expressions such as: "arrival of current spring temperatures", "Length of period with current summer temperatures" or "shift of the arrival of the temperature of current season". I understand that these expressions are not as appealing or as sexy, but they seem more accurate to me. At the very least, if not substituting expressions, the authors should include a discussion on this issue.

Minor questions

lines 16-18, 22-24 and other places through the text:

The summer season will lengthen shouldn't be equal to the sum of the advance in the arrival of summer and the delay in the arrival of autumn?

Section "Shifting seasonality of lakes during the historic period"

How do you explain that spring lake water temperature has arrived earlier in the Northern Hemisphere (lines 95-96) and spring air temperature has arrived earlier in the Southern Hemisphere (lines 107-108)? Same for the delay in arrival of autumn temperatures (109-111 and 113-115).

Line 107.

What data are used to analyse the air temperature? Also ERA 5? It must be indicated in the methodology section.

Lines 194 and 231.

I think that quantifying this difference does not make much sense. Just say that the changes are an order of magnitude greater than the natural fluctuation

Section "Implications of shifting seasons in lakes"

Even without detailing or quantifying, I think that authors should mention and discuss other physical-chemical consequences of climatic shift, namely on:

Thermal stratification and lake turnover;

ice cover regime, one of the most important drivers of ecological change;

water pH

Lake carbon cycle.

Eventually also in lake size, namely in man-made lakes in Mediterranean regions.

Line 381: divide by half the difference in lake temperature

Line 382: following months, in month year-1)

Rui Salgado

Referee #1 (Remarks to the Author):

The manuscript presents an assessment of how the lake surface water temperature is changing in key moments of the year by means of shift seasons in lakes. The pace of shifting seasons in lakes has been calculated for the historic period (using ERA5 data) and for the future using simulations under three different RCPs. Moreover, the author presents the same analysis using satellite spatially distributed data for the Laurentian Great Lakes. The topic of the manuscript is very relevant and interesting, and the global scale of the analysis allows also drawing large-scale conclusions. The topic is in line with the journal scope and the manuscript is well written. I recommend the publication of this study but I think that some aspects should be clarified before publishing the manuscript.

Thank you kindly for reviewing this manuscript. I very much appreciate the reviewer taking the time to carefully consider this work and to provide a number of suggestions for improvement. Please find below responses to the specific reviewer comments.

In the following paragraph, I collect the major and the minor issues from my review.

Thank you again for these comments

Major issues:

- Definition of seasons.

The analysis takes as bases the definition of seasons as following: spring (March-May in Northern Hemisphere and September-November in the Southern Hemisphere), summer (June-August in Northern Hemisphere and December-February in the Southern Hemisphere), and autumn (September-November in the Northern Hemisphere and March-May in Southern Hemisphere). Generally this is accepted, however, some recent studies show that depth plays an important role in defining when the temperature reaches its maximum for example so when summer occurs for example. This is related to the fact that phenology in lakes, is strongly set by stratification as well, not only temperature as in land ecosystems. To take into account this I would suggest to try to do the same analysis considering a moving window of months, meaning to use still 3 months wide window but spanning through the all year to be sure that the real summer, spring and autumn of each lake is properly captured. Ex: JFM, FMA, MAM, ...

Thank you for this suggestion. The reviewer is indeed correct that depth can play an important role on when different “seasons” occur in lakes, and I appreciate their suggestion to define seasons using a moving window approach. Initially, my aim was to present the result in a way in which they would be comparable to marine and terrestrial ecosystems, and also applicable to large-scale patterns rather than to a specific lake type or those situated in a particular part of the world. However, I appreciate that many lentic systems are different and, indeed, that the timing of seasonal events will not always be the same. Thus, I think that including a combination of the “traditional” and “lake-specific” seasons is useful.

In turn, I now include results from the rolling window approach when investigating the pace of shifting seasons in the Great Lakes. By performing this analysis, I now include 6 additional figures (Figs S16-21) and 1 additional table (Table S5) in the supplementary information file.

For completeness, I also present the results using a moving window approach for the global analysis during the historic to contemporary period. These results are shown in Figs S22-S24 and summarised in Table S6. I agree with the reviewer that this is a useful exercise, and a good addition to the manuscript. However, I would prefer to not go into much detail in the main text as I feel this will complicate the story and make it difficult for the reader to follow. Thus, I would prefer to keep a consistent definition in the manuscript whereby I present the results for the “traditional” definition of seasons but include this specific caveat for the Great Lakes - I hope that the reviewer agrees that this is appropriate.

Please also note that when a season is defined as Jul-Sep, the seasonal rate of change is negative (see Fig. S22) throughout most of the Northern Hemisphere, meaning that the lakes have entered the “cooling part” of their seasonal cycle, which suggests that it might not be appropriate to consider as “summer”. Indeed, if we look at Fig. S24, we can see that the seasonal shift is negative during Jul-Sep, which more closely follows the patterns observed (and expected) for autumn.

[Furthermore, I describe in the main text that previous studies have largely shown that maximum NH temperature in lakes occurs in Jun-Aug, which is somewhat expected given that the majority are small/shallow (thus would follow air temperature closely).]

Thank you for this suggestion. I think this section of the paper has now improved considerably as a result and will be directly relevant for those interested in the larger lakes of the world.

- Description of how to calculate the pace of shifting seasons.

The description of how the pace of shifting season has been calculated (from L366 on) it is an important point for the reproducibility of the study. I would suggest to add a step by step description of the calculation method in the supplementary. My main question in this regard would be the following: let’s take spring as an example, April is the central month so you calculated the long term trend for April and then divide this by the difference in temperature between March and May but for each year? Meaning that you end up having one value for each year (1981-2020)? How do you treat then this time series? I suggest adding some information on this point.

Thank you for this suggestion. I now include additional information on how the method is applied. I had not specifically mentioned in the initial draft that the seasonal rate of change is based on a climatology – this information is now provided. Specifically, the method proposed by Burrows et al., (2011), using spring as an example (1980-2021), is as follows:

1. Calculate the long-term trend for April (1980-2021)
2. Calculate the temperature difference between March and May (then divide by 2) for each year of the study (1980-2021).
3. From point 2 above, you calculate the average of these which provides a climatology for the seasonal rate of change
4. Finally, you divide the outcome of point 1 with point 3 above to estimate the pace of the shifting season?

I also now specifically state which function (shiftTime) in the VoCC package in R was used for the calculations (<https://github.com/JorGarMol/VoCC/blob/master/R/shiftTime.R>), which follows the methods introduced by Burrows et al., (2011). In turn, the results presented should be reproducible and the reader can look and use the code available on GitHub.

- Northern and Southern Hemisphere.

The number of lakes or “aggregated lakes” in the two hemisphere could be very different and somehow influence the statistics of the seasonal shifts. I suggest adding information on this.

Yes, the reviewer is correct. The number of representative lakes will differ between the Northern and Southern Hemisphere. This is due to (i) the global distribution of lakes (i.e., there are more lakes in the northern hemisphere), and (ii) the criteria chosen for selecting the study sites when calculating the seasonal metrics (e.g., removing lakes with ice cover). The number of lakes included in each of the calculated metrics is now described in the text, and I mention this caveat and how it can influence any direct comparison of the results.

- Comparison between historic and future pace of shifting seasons.

The comparison with the shifting seasons in air temperature is very interesting and meaningful but I suggest adding a comparison between the historic and future rates. Particularly, I was curious to understand why the historic values reported for example for spring using the historic period (L95) are higher than values for spring using the first 2 RCPs as future scenarios (L189-190). I suggest adding a discussion of the comparison for all seasons.

This is largely due to the specific characteristics of the future emission scenarios. Specifically, RCP 2.6 is a low-emission scenario where emissions start declining at around 2020 (and hence the rate of warming is reduced thereafter), 6.0 (medium-high-emission scenario where emissions peak at around 2080 and then decline), and 8.5 (high-emission scenario where emissions continue to rise throughout the 21st century). So, when calculating the shifting seasons from 2021 to 2099 using RCP 2.6 the rate of change in monthly (e.g., April) water temperature (i.e., the numerator of the equation) will be less than during the historic period (1980-2021) when lakes are experiencing accelerated warming throughout. Similarly, for RCP 6.0 the decrease in greenhouse gas emissions, and thus global warming, toward the year 2080 will influence the calculated rate of change in lake temperature. This information is now included in the main text.

Minor issues:

- L11 – “Much of the focus of....have focused...”. I suggest changing one of the 2 "focus".

Agreed. This has now been changed.

- L11-12 – “mean temperature” means temporal mean in this case right? I suggest to specify because the reader could interpret this as spatial mean otherwise.

Yes, correct. Changed to “temporally averaged “

- L71 – “aggregated typical lake” becomes much more clear after reading the Supporting Information so I suggest adding a reference to that explanation by directly adding part of the explanation in the main text.

Thank you for this suggestion. I have now added a more detailed description in the main text.

- L168 – “later by -5.1”. I suggest changing it to earlier and adding a positive number. This would help the reader.

I appreciate this comment by the reviewer and, indeed, think that it makes sense. However, to be comparable with the climate science literature (e.g., that of Burrows et al), I would prefer to quote these as negative values. Also, I believe that this helps the reader to distinguish between the seasons in the northern/southern hemisphere in the figures – if I changed the values in the text, they would also need to change in the figures, and this would make them less impactful (in my opinion). For now, I have kept these as they were in the original version but, if the reviewer feels strongly about this, I can change make the suggested changes. However, I have now changed the text and removed the phrase ‘later’ and ‘delayed’ when referring to a negative value in autumn, as I appreciate that it is confusing and is not very “clean”.

- Figure 1 – I suggest adding “spring” on the left of panel a and “autumn” on the left of panel c to have the figure more self-explicative.

Yes, good idea. Figure changed.

- Figure 4 – I suggest using “summer lengthening” instead of “longer summer” for the colorbar and consistently in the manuscript. Also, the abbreviations NH and SH should be reported in the caption.

Agreed. Changes made.

- Table S1 – I suggest adding the analogous table with the future scenarios. I know that values for single lakes are reported in tables from S5 on but having an aggregated table for the Northern and Southern Hemispheres would help the comparison with the historic periods.

Agreed. Now included.

Referee #2 (Remarks to the Author):

Woolway analyzed past (modeled and observed) and future (modeled) lake surface water temperature data to assess the pace of shifting seasons in lakes worldwide. The analysis has been done considering three seasons (spring, summer, and autumn) and different climate change scenarios (RCP 2.6, 6.0, 8.5 and pre-industrial control simulation). The present study follows the line of recent studies published by the Author and dealing with the effects of climate change on lakes' temperature and stratification phenology, e.g.: Woolway and Merchant (2019, <https://doi.org/10.1038/s41561-019-0322-x>), Woolway and Maberly (2020, <https://doi.org/10.1038/s41558-020-0889-7>); Woolway et al. (2021, <https://doi.org/10.1002/lol2.10231>); Woolway et al. (2021, <https://doi.org/10.1038/s41467-021-22657-4>); Woolway et al. (2022, <https://doi.org/10.1093/biosci/biac052>); Woolway et al. (2022, <https://doi.org/10.1029/2021GL097031>).

The methodological approach used by the Author is the one proposed by Burrow et al. (2011) to assess the pace of shifting climate in marine and terrestrial ecosystems. Burrow et al. (2011) proposed two indicators of climate change shift: the velocity of climate change and the seasonal climate shift. The first indicator has been used by the Author in a previous manuscript (Woolway and Maberly, 2020 <https://doi.org/10.1038/s41558-020-0889-7>). The latter has been used in the present study. I recognize that the Author is brilliant in presenting worldwide trends and patterns using original methodologies or applying existing ones, as in this case. However, I also think that the key message of this manuscript is not new compared to that of the previous studies by the Author, but it is the same presented in a different way.

Thank you for reviewing this article. I truly appreciate the reviewer's comments. I think that it is also important to highlight that the papers listed above are different from the present study. I understand that these previous studies have a common theme, in that they investigate climate change responses in lakes, but the topic of the present study is different, and they focus on very different responses to global warming. I have now highlighted the novel contribution of this study in the main text.

Importantly, I noticed that section "Implications of shifting seasons in lakes" is made of several sentences almost entirely copied-pasted from previous manuscripts of the Authors (Woolway et al., 2021, <https://doi.org/10.1038/s41467-021-22657-4>; Woolway et al., 2022, <https://doi.org/10.1093/biosci/biac052>; Woolway et al., 2022, <https://doi.org/10.1029/2021GL097031>), where the key words of these previous studies (e.g., heatwaves, stratification phenology) have been substituted with the one of the present study (i.e., seasonal shift). This reinforces the idea that the message of this study is not entirely new. This is all the more true when we consider that the literature on the topic is quite vast and increasing at a fast pace, e.g., Jane et al (2022, <https://doi.org/10.1111/gcb.16525>), Oleksy and Richardson (2021, <https://doi.org/10.1029/2020GL090959>), Kraemer et al. (2021, <https://doi.org/10.1038/s41558-021-01060-3>), Dokulil et al (2021, <https://doi.org/10.1007/s10584-021-03085-1>), Piccolroaz et al. (2021, <https://doi.org/10.1016/j.ejrh.2021.100780>), Răman Vinnå et al. (2021, <https://doi.org/10.1038/s43247-021-00106-w>), Shatwell et al. (2019, <https://doi.org/10.5194/hess-23-1533-2019>), Niedrist et al. (2018, <https://doi.org/10.1007/s10584-018-2328-6>), Butcher et al. (2015, <https://doi.org/10.1007/s10584-015-1326-1>)

I think that the reviewer's comments about the discussion is fair. I have now modified the discussion section accordingly, and now include further information highlighting the novelty of this work in the introduction and elsewhere in the main text.

As described, the key novelty of this work is in the use of a methodological approach to quantify changes to the timing of the seasons in lakes. This has not been done before, and no other study has considered changes to the timing of the seasons in lakes, which is critical for many aquatic

ecosystem processes. Many studies have focused on climate change impacts in lakes, but this is a very broad topic as I'm sure the reviewer will appreciate. For example, some of the papers listed by the reviewer above focus on very different points to those highlighted in this paper – e.g., the Jane et al., study investigates lake oxygen concentrations.

As for the methodology:

- as correctly pointed out by the Author, LSWT seasonality is largely influenced by seasonality of surface heat fluxes, timing of ice-cover and lake depth. I think that the approach used by the Author (FigS1) at the global scale can seriously hide these effects, resulting in a flattening (simplification) of the results in terms of deep understanding/representation of the processes and of the inter-lake variability. As an example, the three seasons investigated by the Author correspond to very different periods of the LSWT annual cycle of the Great Lakes (see e.g., Fig 1 in Cannon et al., 2019, <https://doi.org/10.1029/2019GL082916>). This certainly adds complexity to the processes and to the analysis.

As the reviewer suggests, depth can certainly play an important role on when different “seasons” occur in lakes. Initially, my aim was to present the result in a way in which they would be comparable to marine and terrestrial ecosystems, and also applicable to large-scale patterns rather than to a specific lake type or those situated in a particular part of the world. However, I appreciate that many lentic systems are different and, indeed, that the timing of seasonal events will not always be the same. Thus, I now include a combination of “traditional” and “lake-specific” seasons in this study. Specifically, I now include results showing the seasonal shift if summer (for example) is defined as Jul-Sep (and so on), following a rolling window approach, as suggested by reviewer 1. Most notably, I apply this method for the Great Lakes, where the seasonality is known to be different. By performing this analysis, I now include an additional 9 figures and 2 tables in the supplementary information file. Note that I would prefer to keep a consistent definition in the manuscript whereby I present the results for the “traditional” definition of seasons but include this specific caveat for the Great Lakes.

For completeness, I also present the results using a moving window approach mentioned above for the global analysis during the historic to contemporary period. These results are shown in Figs S22-S24 and summarised in Table S6. However, please note that when a season is defined as Jul-Sep, the seasonal rate of change is negative (see Fig. S22) throughout most of the Northern Hemisphere, meaning that the lakes have entered the “cooling part” of their seasonal cycle, which suggests that it might not be appropriate to consider as “summer”. Indeed, if we look at Fig. S24, we can see that the seasonal shift is negative during Jul-Sep, which more closely follows the patterns observed (and expected) for autumn.

I now also cite the work mentioned by the reviewer and also highlight the complexity of this process across lakes, and that it should be considered when interpreting the results.

I also agree with the reviewer that large scale studies such as this often mask local scale patterns of variability which can also be seen clearly in, for example, studies of global air temperature change. However, such global-scale studies are essential for understanding Earth System responses to climate change.

- Fig1 b and d: showing NH and SH with different colors would be useful to appreciate to what extent the bimodal distribution is explained by latitude above/below the equator.

Yes, agreed. This figure has now been modified following the reviewer's suggestion.

Minor comments:

- L11: "focus", "focused"

Sentence changed.

- L13: "event" to "even"

Yes. Thank you.

- FigS1: this is exactly as in Burrow et al. (2011), except for minor details. I would not include June and write that the schematic applies for spring.

Changes made. Note that in the caption we state that this is re-drawn from Burrows et al., (2011). We have now elaborated on the method, as requested by one of the other reviewers.

- FigS6, L690 remove "are"

Corrected.

Referee #3 (Remarks to the Author)

The submitted manuscript reports a quality work carried out with the aim of filling the knowledge gap about present and future climatic shifts in lakes. The work is of great topicality and significance to the field. The manuscript is original, very well organized, well written and clear.

Thank you very much for reviewing this manuscript. I greatly appreciate the time taken to provide a detailed review. Thank you also for the suggestions on how to improve this work. This is much appreciated.

Major Issue

I have a single fundamental question about the work and it was because of this issue that I took longer than I intended to review the article. My apologies to the authors and editors. The question is about the definition of shifting seasons and how to quantify it. I followed the references and understood that it is a methodology already widely used in studies about climatic shifts in both marine and terrestrial ecosystems. I recognize that the measure of shifts in temperature, as done in the paper, can be a good quantifier of climate change, but it seems to me a bit abusive to use it as a synonym for "seasonal shift". The concept of season in climate is much more than the recorded temperature. To be more precise, in my opinion what this parameter measures is the shift of the arrival of the temperature of current season. This measure can be seen also as an indicator of the increase in temperature of the season, and not necessarily of a shift of the season. This problem is likely to be magnified in summer as, namely at high latitudes, in the Northern Hemisphere July is the hottest month so the differences between August and June temperatures are small. This question came to me sharply when I analysed the results of the study. For example, when presenting the estimate that the summer season will lengthen at a rate of 12.1-days decade⁻¹ (line 24), means that it is estimated that the duration of the summer will increase by about 100 days at the end of the century, that is, it will last for more than two seasons. In fact, what the study actually indicates is that, at the end of the century, there will be another 100 days with current summer temperatures. This conclusion is, by itself, very interesting and important, but it doesn't seem to me to be the same as saying that summer is twice as long.

Yes, I think this is a fair and valid point by the reviewer. I tend to agree that it is more accurate to write the calculated metric as the "shift in the arrival of the temperature of the current season", despite it not being as appealing (as the reviewer describes above/below). I have now modified the paper following the reviewer's suggestion. Specifically, I now explicitly state in the paper that, for example, by the end of the century there will be an additional xx days of summer, relative to current temperatures. In addition, I now state that the arrival of the seasons (spring/autumn), relative to historic temperatures, will change by yy days. This comment, as well as the one below, mostly apply to the future simulations as, intuitively, "current temperatures" are inherent in the definition of the historic to contemporary period. Note that I have not changed the title, as I believe this needs to be attractive (and brief) to a broad audience.

I admit that it can be a preciousness, but I would like to suggest that authors pay attention to this issue and mitigate the use of the terms like "season shift", "arrival of spring", "length of Summer". Instead, it seems more appropriate to use expressions such as: "arrival of current spring temperatures", "Length of period with current summer temperatures" or "shift of the arrival of the temperature of current season". I understand that these expressions are not as appealing or as sexy, but they seem more accurate to me. At the very least, if not substituting expressions, the authors should include a discussion on this issue.

This is again a valid point by the reviewer. These changes have been made throughout the manuscript and, where applicable, we have changed "arrival of spring" to "arrival of current spring temperatures", for example.

Minor questions

The summer season will lengthen shouldn't be equal to the sum of the advance in the arrival of summer and the delay in the arrival of autumn?

Yes, this is a good point and would certainly be the case if the same number of lakes had information during the different seasons being compared (e.g., summer and autumn). However, due to the presence of ice cover and a small climatological seasonal rate of change, some lakes are included in the calculation for autumn temperatures but not summer temperatures (and vice versa). I now provide further information of this caveat in the main text, as well as provide information on the number of lakes included in each of the summary statistics.

Section "Shifting seasonality of lakes during the historic period"

How do you explain that spring lake water temperature has arrived earlier in the Northern Hemisphere (lines 95-96) and spring air temperature has arrived earlier in the Southern Hemisphere (lines 107-108)? Same for the delay in arrival of autumn temperatures (109-111 and 113-115).

Line 107.

The explanation here follows the comments made by the reviewer above. Notably, as mean temperatures rise, the seasons threshold from the previous year(s) is exceeded more quickly hence the advance/delay in the arrival of the seasons. The methodological figure in the supplementary file (Fig. S1) should help with understanding this principle. I hope that I have understood (and thus answered) the question here?

What data are used to analyse the air temperature? Also ERA 5? It must be indicated in the methodology section.

Yes, also ERA5. This information was included in the second paragraph of the main text, but I now also include this in the methods. Thank you for bringing this to my attention.

Lines 194 and 231.

I think that quantifying this difference does not make much sense. Just say that the changes are an order of magnitude greater than the natural fluctuation

OK. Changes made.

Section "Implications of shifting seasons in lakes"

Even without detailing or quantifying, I think that authors should mention and discuss other physical-chemical consequences of climatic shift, namely on:

Thermal stratification and lake turnover;

ice cover regime, one of the most important drivers of ecological change;

water pH

Lake carbon cycle.

Eventually also in lake size, namely in man-made lakes in Mediterranean regions.

Thank you for these suggestions. I now mentioned these other physical-chemical consequences of climate change in the last paragraph.

Line 381: divide by half the difference in lake temperature

Line 382: following months, in month year⁻¹)

Thank you. These sentences have been modified.

Rui Salgado

Reviewer #1 (Remarks to the Author):

The pace of shifting seasons in lakes.

All my previous suggestions have been implemented in a very clear way and the manuscript has improved. I only collected minor suggestions for the author which might be worth considering before publication:

- L19-20: "This study also suggests that future alterations in the timing of seasonal temperatures will be even greater." This is strongly dependent on which RPC you consider. For spring and summer NH for example this is true only in the case of RCP 8.5. This is actually well noted at lines 270-271 of the manuscript where you say: "Note that in some instances, the estimated pace of the shifting seasons is faster during the historic (1980-2021) than the future (2021-2099) period, notably under RCP 2.6.". I suggest changing the sentence in the abstract accordingly.
- One clarification about the analysis of satellite data for the Great Lakes. In the case of the analysis of the satellite data it would be useful to know if the ERA5 air data have been considered per pixel or as a unique time series for the all lake? In case of per-pixel air temperature, how did you solve the problem of different spatial resolution to the GLSEA lake surface water temperature data? I suggest adding this in the method for reproducibility.
- Fig S9-S10. Each panel reflects a range of 3 months but only 1 month is mentioned at the upper right of the picture. I suggest changing labels with the full range to avoid confusion.
- Tables. I suggest reporting only 1 digit after comma like in the text, this should reflect better the accuracy.
- Table S7-S9. I suggest adding a row with the historic to contemporary period (1980-2021). This would help the reader to compare the values in an easier way.

Reviewer #2 (Remarks to the Author):

I believe that this manuscript is well written but I confirm that I am a bit hesitant to consider it as representing "important advances of significance", considering the previous studies on the topic. I recognize that the methodology used by the author is new (in the context of limnology), but several other previous studies analyzed the consequences of climate change on stratification phenology, duration and intensity, which is another and I think more appropriate way to see seasons in lakes.

I appreciate the revision of the discussion and the addition of a paragraph on the use of a moving window approach, both were needed.

As for this latter point, the results indicate that quantifying the shifts in the timing of seasons is largely dependent on the definition of the averaging window (Table S5), as it is expected to be considering the major role of stratification (depth) on lakes dynamics. This may substantially bias the quantification of the pace of shifting seasons in lakes presented by the author in his global application. Caution should be used in interpreting these results to avoid misinterpretations.

Reviewer #3 (Remarks to the Author):

The author answered very satisfactorily to all the questions I asked in the first review and took my suggestions into account.

As I see it, the author has also responded well to the other reviewers.

The submitted manuscript reports a quality work carried out with the aim of filling the knowledge gap about present and future climatic shifts in lakes. The work is of great topicality and significance to the field. The manuscript is original, very well organized, well written and clear. The English in the article seems to me to be of very good quality, certainly much better than mine, so I dare not suggest any language corrections.

In my opinion, the manuscript can be published as it is in Nature Communications.

Referee #1 (Remarks to the Author):

All my previous suggestions have been implemented in a very clear way and the manuscript has improved. I only collected minor suggestions for the author which might be worth considering before publication:

Thank you again for reviewing this manuscript – and thank you for the additional suggestions outlined below.

- L19-20: “This study also suggests that future alterations in the timing of seasonal temperatures will be even greater.” This is strongly dependent on which RPC you consider. For spring and summer NH for example this is true only in the case of RCP 8.5. This is actually well noted at lines 270-271 of the manuscript where you say: “Note that in some instances, the estimated pace of the shifting seasons is faster during the historic (1980-2021) than the future (2021-2099) period, notably under RCP 2.6.”. I suggest changing the sentence in the abstract accordingly.

Thank you for this suggestion. The abstract has now been modified accordingly. Notably, I have now included the word “could” instead of “will”. The distinction between the low and high RCP scenarios are also included in the abstract. Also, please note that the abstract has been shortened to 150 words, in line with the journal requirements.

- One clarification about the analysis of satellite data for the Great Lakes. In the case of the analysis of the satellite data it would be useful to know if the ERA5 air data have been considered per pixel or as a unique time series for the all lake? In case of per-pixel air temperature, how did you solve the problem of different spatial resolution to the GLSEA lake surface water temperature data? I suggest adding this in the method for reproducibility.

In this study I did not compare the GLSEA data with air temperatures from ERA5 (as the reviewer suggests, they are not directly comparable given the different spatial resolution). The comparison with air temperature is made at the larger hemispheric scale where the grid cells between lake surface temperature and air temperature are consistent.

- Fig S9-S10. Each panel reflects a range of 3 months but only 1 month is mentioned at the upper right of the picture. I suggest changing labels with the full range to avoid confusion.

Agreed. Now added.

- Tables. I suggest reporting only 1 digit after comma like in the text, this should reflect better the accuracy.

Agreed. Changed.

- Table S7-S9. I suggest adding a row with the historic to contemporary period (1980-2021). This would help the reader to compare the values in an easier way.

Given that the historic simulations within ISIMIP2b end in 2005 (they are forced by RCP scenarios thereafter), it is not possible to calculate the seasonal rates for the period 1980-2021. Therefore, the ERA5 data are used for assessing changes during the period 1980-2021, as included in the other supplementary tables.

Referee #2 (Remarks to the Author):

I believe that this manuscript is well written but I confirm that I am a bit hesitant to consider it as representing "important advances of significance", considering the previous studies on the topic. I recognize that the methodology used by the author is new (in the context of limnology), but several other previous studies analyzed the consequences of climate change on stratification phenology, duration and intensity, which is another and I think more appropriate way to see seasons in lakes. I appreciate the revision of the discussion and the addition of a paragraph on the use of a moving window approach, both were needed. As for this latter point, the results indicate that quantifying the shifts in the timing of seasons is largely dependent on the definition of the averaging window (Table S5), as it is expected to be considering the major role of stratification (depth) on lakes dynamics. This may substantially bias the quantification of the pace of shifting seasons in lakes presented by the author in his global application. Caution should be used in interpreting these results to avoid misinterpretations.

Thank you again for reviewing the article. In the revised manuscript, I now include a caveat that there might be a bias in the results towards how the definition of the averaging window has an important impact on quantifying the shifts in the timing of seasons. I also take extra care to assess the study in the light of other, already published literature – this involves citing all relevant papers that the reviewer mentioned in their previous assessment of the work. Although, please note that the reference list is already exhaustive.

Referee #3 (Remarks to the Author)

The author answered very satisfactorily to all the questions I asked in the first review and took my suggestions into account. As I see it, the author has also responded well to the other reviewers.

The submitted manuscript reports a quality work carried out with the aim of filling the knowledge gap about present and future climatic shifts in lakes. The work is of great topicality and significance to the field. The manuscript is original, very well organized, well written and clear. The English in the article seems to me to be of very good quality, certainly much better than mine, so I dare not suggest any language corrections.

In my opinion, the manuscript can be published as it is in Nature Communications.

Thank you for your positive assessment of the manuscript.